# Risk Factors for Pressure Injuries in Adult Patients: A Narrative Synthesis

**DOI:** 10.3390/ijerph19020761

**Published:** 2022-01-11

**Authors:** Man-Long Chung, Manuel Widdel, Julian Kirchhoff, Julia Sellin, Mohieddine Jelali, Franziska Geiser, Martin Mücke, Rupert Conrad

**Affiliations:** 1Department of Psychosomatic Medicine and Psychotherapy, University Hospital Bonn, 53127 Bonn, Germany; Franziska.Geiser@ukbonn.de (F.G.); Rupert.Conrad@ukbonn.de (R.C.); 2Institute of Product Development and Engineering De sign, Technische Hochschule Köln, 50679 Cologne, Germany; manuel_widdel@yahoo.de (M.W.); Julian.Kirchhoff@icloud.com (J.K.); mohieddine.jelali@th-koeln.de (M.J.); 3Department of Digitalization and General Practice, University Hospital RWTH Aachen, 52074 Aachen, Germany; Julia.Sellin@ukbonn.de (J.S.); Martin.Muecke@ukbonn.de (M.M.)

**Keywords:** narrative synthesis, pressure injury, prevention, treatment, risk factors, systematic review

## Abstract

Pressure injuries remain a serious health complication for patients and nursing staff. Evidence from the past decade has not been analysed through narrative synthesis yet. PubMed, Embase, CINAHL Complete, Web of Science, Cochrane Library, and other reviews/sources were screened. Risk of bias was evaluated using a slightly modified QUIPS tool. Risk factor domains were used to assign (non)statistically independent risk factors. Hence, 67 studies with 679,660 patients were included. In low to moderate risk of bias studies, non-blanchable erythema reliably predicted pressure injury stage 2. Factors influencing mechanical boundary conditions, e.g., higher interface pressure or BMI < 18.5, as well as factors affecting interindividual susceptibility (male sex, older age, anemia, hypoalbuminemia, diabetes, hypotension, low physical activity, existing pressure injuries) and treatment-related aspects, such as length of stay in intensive care units, were identified as possible risk factors for pressure injury development. Health care professionals’ evidence-based knowledge of above-mentioned risk factors is vital to ensure optimal prevention and/or treatment. Openly accessible risk factors, e.g., sex, age, BMI, pre-existing diabetes, and non-blanchable erythema, can serve as yellow flags for pressure injury development. Close communication concerning further risk factors, e.g., anemia, hypoalbuminemia, or low physical activity, may optimize prevention and/or treatment. Further high-quality evidence is warranted.

## 1. Introduction

Pressure injuries or bedsores have deleterious implications for the patient, resulting in enormous suffering, diminished quality of life, and higher mortality [1]. A systematic assessment of mortality in the context of the Global Burden of Diseases Study found that pressure injuries led to 20,300 deaths worldwide in 2017 [2] compared to 13,700 deaths in 1990 [3]. Previous research highlighted mobility/activity, perfusion, e.g., diabetes, and skin/pressure injury status as independently predictive of pressure injury development [4] while among critical care patients, age, mobility/activity, perfusion, and vasopressor infusion were statistically significant in multivariate analysis [5]. Those risk factors can be categorized into two domains as postulated by an updated conceptual framework [6]: “mechanical boundary condition”, e.g., mechanical load, shear, pressure, etc., and “susceptibility and tolerance of the individual”, e.g., individual physiology, geometry of tissues, etc. Risk factors connected to the second domain “susceptibility and tolerance of the individual” seem to be more relevant than those connected to the first. However, statistical independence does not imply causality as the results achieved in multivariate analysis are influenced by other risk factors included in the same statistical model which can vary between studies. The above mentioned findings on the basis of the systematic reviews by Coleman et al. [4] and Alderden et al. [5] highlight the connectivity and interplay of the various risk factors that could lead to pressure injury development. The first review systematically investigated risk factors for pressure injuries in adult patients across all settings covering the relevant literature up to March 2010 [4] while the second review focusing on critical care patients included the published research up to December 2016 [5], so relevant evidence from the last decade has not been analysed systematically by means of narrative synthesis yet. The knowledge of evidence-based risk factors for the development of pressure injuries is highly relevant to nursing practice with implications for prevention, diagnostic assessment, and nursing interventions.

Against this backdrop, a comprehensive narrative synthesis of risk factors for pressure injury was performed based on the systematic and rigorous analysis of the published research from inception to December 2020.

## 2. Materials and Methods

A narrative synthesis regarding various risk factors for pressure injuries was performed for heterogenous studies. This work followed the guidelines of “Preferred Reporting Items for Systematic Reviews and Meta-Analyses” [7] (PRISMA; Appendix A). Inclusion and exclusion criteria were based upon the PICOS criteria without the inclusion of “comparator/control” as the included study designs do not always utilize control groups or similar [8,9]. Defining inclusion criteria for appropriate studies based on control groups would have excluded relevant study designs for risk factor research and was therefore omitted. An additional meta-analysis with prominent demographic characteristics age and sex and the widely used assessment scale, the Braden scale with its subscales, was also conducted which will be presented separately.

Inclusion Criteria: (a) adult patients (>18 years old) in any settings, (b) *n* ≥ 200, (c) any physiotherapeutic or medical treatment/intervention or implementation of medical devices, e.g., mattresses or foam dressing regarding pressure injury prevention, (d) main outcome: development of pressure injury stage ≥ 1 during the study period, (e) primary research, (f) prospective cohort studies/retrospective record reviews/randomized controlled trials (RCT), (g) use of multivariate analysis. Exclusion Criteria: (a) cross-sectional studies/prevalence studies/case studies/other systematic review or meta-analysis, (b) language other than German or English.

Ethical approval from an ethics committee was not required as this review was based on already published studies. A study protocol was registered on the website PROSPERO under the following ID CRD42020215378.

### 2.1. Data Collection

The following electronic databases were searched: PubMed, Embase, CINAHL Complete, Web of Science and Cochrane Library. The search strategy, which consisted of two synonym groups, was first used from inception until end of January 2020 and a second time from end of January 2020 until mid of December 2020. One of the groups consisted of synonyms regarding “pressure injury” and the other about “risk factors”. MeSH terms were appropriately utilized in PubMed. An adjusted search strategy was used for each electronic database, further details are displayed in “Appendix A”. Other research papers were additionally identified through forward and backward chaining as well as internet search and other reviews. Search results were imported to EndNote where deduplication took place. Titles and abstracts were screened and checked against the inclusion and exclusion criteria by two reviewers (M.-L.C. and R.C.), while discrepancies were dealt with through discussion and consensus with other review authors (M.W., J.K., J.S., M.J., F.G. and M.M.). Full copies of the included studies were acquired and checked if they met the criteria.

### 2.2. Data Extraction

Three review authors (M.-L.C., M.M. and R.C.) extracted data from each study independently using a standard form. The extracted data included information regarding the study setting, study population, study design and statistical analysis method, numbers of newly developed pressure injuries during the study period, grading cutoff as outcome, effect sizes (odds ratios (OR), hazard ratios (HR)), and their respective confidence interval (Appendix A).

### 2.3. Risk of Bias

To assess the risk of bias for each study, a slightly modified version of the QUIPS tool was used [10]. Three review authors (M.-L.C., M.M. and R.C.) independently assessed the risk of bias for each included study which was checked by other review authors (J.S., M.J. and F.G.). Disagreements were resolved through discussion and consensus. The QUIPS tool consists of six main domains: “study participation”, “study attrition”, “prognostic factor measurement”, “outcome measurement”, “study confounding”, and “statistical analysis and reporting”. Each domain can be evaluated as low, moderate, or high.

In order to have an overall risk of bias for each study, we took a similar approach to another systematic review [11]. The following four domains, “study participation”, “study attrition”, “study confounding”, and “statistical analysis and reporting”, were designated as key domains by the authors, where their evaluation carries more weight to the overall evaluation. Since the domain “study attrition” could not be applied to every included study in this review, we decided to transfer its key domain-status to “outcome measurement” as the accurate and reliable assessment and identification of pressure injuries is of utmost importance [5].

Low risk of bias was present if all four key domains had low risk or, at most, only one key domain had moderate risk of bias. If one of the remaining two domains (“study attrition” and “prognostic factor measurement”) was at least evaluated as high risk of bias, however, the overall evaluation would be downgraded to moderate risk of bias. Moderate risk was also present if not more than one key domain is assessed as high risk or two key domains as moderate risk. High risk was present if at least two key domains had high risk or at least three key domains had moderate risk of bias. The following domains were assessed for each study:

#### 2.3.1. Study Participation

This domain is rated based on the included study sample. Factors such as adequate description of source population and baseline study sample, details about recruitment and time frame as well as the applied inclusion and exclusion criteria were assessed.

#### 2.3.2. Study Attrition

The majority of included studies were not RCTs with a defined follow up which resulted in a very low assessment rate for this domain. Evaluation was based on adequate response rates (≥80%) [12] and if the reasons for a potential loss to follow up were provided as well as the characteristics of patients that were lost to follow up.

#### 2.3.3. Prognostic Factor Measurement

This domain focused on the assessed prognostic factors and their measurement method. Continuous variables or appropriate cut-offs should be reported and the measurement method is the same across all settings and among participants. This review considered each prognostic factor that is ultimately included in a multivariate analysis, which leads to on overall assessment for the sum of prognostic factors instead of a singular factor.

#### 2.3.4. Outcome Measurement

A clear definition of the outcome has to be provided for a favorable rating, in this case a newly developed pressure injury stage 1 or higher according to EPUAP/NPIAP and PPPIA guidelines or equivalent, e.g., ICD 10 [13,14]. Ideally, the identification and grading of newly developed pressure injuries were evaluated by an experienced nurse or staff members that were specifically educated and trained for it.

#### 2.3.5. Study Confounding

The rating is based on the consideration of confounding variables. Like the domain “prognostic factor measurement”, the definition and measurement method of each confounding variable must be sufficiently reported, and they must be accounted for in the study design and statistical analysis. Since common confounding variables (e.g., sex, age) also serve as potential prognostic factors here, the rating for this domain and “prognostic factor measurement” is similar.

#### 2.3.6. Statistical Analysis and Reporting

The analytic approach and presentation of data was assessed. Independent prognostic factors should be identified through multivariate analysis, e.g., logistic regression. Additionally, emphasis was put on the “one in ten” rule regarding the reliability of the prediction of each factor [15,16,17]. For each predictor included, there must be at least ten events present. In this case, there must be at least ten patients with newly developed pressure injuries for each included predictor. If this rule of thumb is not satisfied, this domain was rated as high risk of bias automatically.

### 2.4. Data Synthesis

In line with similar systematic reviews [4,5], for narrative synthesis, each risk factor was categorized into domains and subdomains, structured along a conceptual framework [6], as well as separated according to pressure injury outcome stage ≥ 1 or stage ≥ 2 and its statistical (non)significance.

## 3. Results

### 3.1. Study Characteristics

The search strategy was implemented in end of January 2020 and beginning of December 2020 with overall 29,382 search results (Figure 1). Together with 38 studies added through other reviews and sources, deduplication took place with 29,420 studies where, ultimately, 11,517 results remained. Screening of titles and abstracts was done with 11,517 entries where 123 studies were deemed appropriate for full-text analysis. In the end, 67 studies were included in narrative synthesis, with 32 prospective cohort studies [18,19,20,21,22,23,24,25,26,27,28,29,30,31,32,33,34,35,36,37,38,39,40,41,42,43,44,45,46,47,48,49], 26 retrospective record reviews [50,51,52,53,54,55,56,57,58,59,60,61,62,63,64,65,66,67,68,69,70,71,72,73,74,75], and nine RCTs [76,77,78,79,80,81,82,83,84] with a total sample of 679,660.

### 3.2. Pressure Injury Outcome

Thirty-eight studies had pressure injury stage 1 or higher as their main outcome [18,21,23,25,27,29,30,31,32,33,34,35,36,39,40,44,45,46,47,48,50,55,57,58,59,61,67,68,69,70,72,74,75,76,78,79,80,83], 19 studies defined pressure injury stage 2 or higher as their main outcome [19,20,22,26,28,37,38,42,49,51,53,62,63,65,66,73,81,82,84], three studies considered both in their analysis [24,54,77], while seven studies did not specify the stage [41,43,52,56,60,64,71].

Forty-five studies analysed pressure injury development as a dichotomous outcome [18,20,21,23,24,26,27,29,30,32,33,35,36,39,41,42,43,44,45,46,47,48,49,50,53,54,55,56,57,58,60,61,62,63,65,66,68,71,72,74,75,77,79,82,83], 20 studies reported time to the development of pressure injuries [19,22,25,28,31,34,37,38,40,51,52,59,64,67,70,73,76,78,81,84], and two study considered both in their analysis [69,80].

Fourteen studies reported more than one multivariate model in their statistical analysis. The selection of a primary model in these studies was based on the criteria defined by Coleman et al. [4]: main outcome new pressure injury stage ≥ 1, main outcome development of new pressure injuries, model with the widest range of risk factors, total sample or largest sub-group, largest number of newly developed pressure injury, and models without time dependent variables.

### 3.3. Risk of Bias

Of all included studies, 22 studies [21,23,26,35,37,40,54,55,57,58,59,62,63,65,66,67,68,70,72,77,82,83] (33%) had an overall low risk of bias, 29 studies [19,20,22,24,25,27,28,30,31,33,38,39,42,44,46,47,48,49,50,51,53,60,61,73,74,78,80,81,84] (43%) had an overall moderate risk, and 16 studies [18,29,32,34,36,41,43,45,52,56,64,69,71,75,76,79] (24%) had an overall high risk of bias (Figure 2).

Over 90% of studies had a low risk of bias concerning “study participation” because the majority reported sufficient information about their study sample and recruitment, data acquisition, and time frame. “Study attrition” was only applicable for RCTs, where around 50% had low risk of bias and the other 50% had moderate risk of bias, the latter being the result of missing information about dropouts. The domains “prognostic factor measurement” and “study confounding” share similar results with around 75–80% of studies being low risk of bias. Since confounding variables, e.g., age or sex, are also potential risk factors, both domains were assessed similarly. Lack of description of risk factors or categorization/dichotomisation of continuous variables resulted in a lower assessment [85].

The focus of “outcome measurement” was the assessment of pressure injuries by skilled or trained health worker, e.g., nurses, doctors etc., and the definition of pressure injuries by certain and clearly defined standards. Many studies failed to report an adequate staging or assessment of pressure injury by trained health workers, specific pressure injury definition, or both, which resulted in 40% of studies having moderate [18,19,24,25,29,31,32,33,39,42,45,47,51,53,55,57,58,59,62,63,69,72,73,74,76,78,82] and 19% having high risk of bias in this domain [18,22,34,36,41,43,52,56,60,64,71,75,79].

Over 50% of studies had a high risk of bias in the domain “statistical analysis and reporting”. This rating is mostly due to the lack of pressure injury events compared to the amount of risk factors considered in multivariate analysis [15,16,17]. Other reasons were insufficient strategy for model building and selective reporting of results, e.g., no confidence intervals.

### 3.4. Risk Factor Domains

Fifty-eight studies (87%) reported both the included risk factors in the multivariate model as well as the number of risk factors that were independently predictive of pressure injury. Nine studies (13%) only reported risk factors that emerged as statistically significant. Significant as well as non-significant risk factors, if available, from multivariate analysis were summarized and displayed separately for stage ≥ 1 and stage ≥ 2 pressure injury.

#### 3.4.1. Domain 1: Mechanical Boundary Conditions

Risk factors that affect the magnitude and duration of mechanical load as well as the type of loading are assigned to domain 1 (Appendix A). Variables that cause or influence immobility, one of the direct causal factors associated with pressure injury development, were included in this domain along with poor sensory perception and response. Body size was also included here as it can influence the magnitude of mechanical load on the individual [5].

Body size

Variables related to body size e.g., body mass index (BMI), weight and height were entered in 17 studies [18,19,24,26,30,34,43,49,50,53,56,61,62,64,77,83,84]. Out of 23 entered variables, eight emerged as significant predictors of pressure injury development. Two moderate risk of bias studies imply that low BMI contributes to the development of pressure injuries stage ≥ 1 [30,50] whereas one moderate [61] and one high risk of bias study [56] attribute the development to high BMI. Lower body mass and greater weight were also independently associated with pressure injury development stage ≥ 1 in one low [83] and high risk of bias study [18], respectively. For stage ≥ 2 pressure injuries, low BMI was a significant predictor in one low risk of bias study [62] and decreased body weight in one moderate risk of bias study [19].

2.Friction/shear

Eleven studies included friction/shear variables in their multivariate analysis [28,33,40,46,51,54,61,70,73,77,84]. Most of them utilized the friction/shear subscale of the Braden scale. Further, 50% of the included variables yielded a significant result while more low risk of bias studies reported a non-significant result [54,70,77].

3.Interface pressure

Interface pressure refers to pressure that is applied through specific surfaces, like mattresses or foaming, that are used to prevent pressure injuries, but also to pressure that arises from external forces, e.g., use of restrictions, application of force, or surgery positions. The authors of 16 studies included variables related to interface pressure [20,28,30,37,42,44,45,56,59,64,72,77,78,80,82,83]. Overall, 58% of them showed significant results, with the majority of analysed mattresses or foam dressings serving as a protective factor.

4.Subdomain Immobility

Mobility

Fifty-one mobility related variables were analysed in 28 studies [25,26,28,29,30,33,36,40,41,42,44,46,51,54,55,57,61,62,64,65,69,70,73,74,76,77,79,84]. Mobility refers to the ability of individuals to move, adjust, and control their body positions [86]. Among these variables are mobility subscales from risk assessment scales like the Braden or Norton scale, surgery related variables, e.g., waiting time or anesthesia, or activities of daily living (ADL) variables. Altogether, 51% of mobility related variables reached significance in multivariate analysis.

Activity

Authors of 29 studies [19,22,26,28,29,30,31,33,36,38,40,41,46,47,51,56,58,59,61,63,64,69,70,73,76,77,79,82,84] considered 46 variables related to activity in their statistical analysis, which included activity subscales from risk assessment scales and ADL. Around 43% of the included activity related variables were significant predictors of pressure injury development.

5.Sensory perception

Sensory perception as assessed by the Braden subscale was analysed and included in nine studies [33,40,46,54,61,70,73,77,84], where most studies (69%) did not report it as a significant predictor.

6.Mental status/neurological disorders

Variables that describe the mental status, consciousness, or include neurological disorders were entered into the statistical model of 18 studies [20,21,22,26,28,29,34,36,40,41,53,57,58,64,66,68,77,83]. Overall, 29 variables were analysed and 10 of them (34%) emerged as independent predictors of pressure injury development.

7.Turning and repositioning

Only one moderate risk of bias study included turning and repositioning as a potential predictor in its analysis [81]. The implementation rate of the turning schedule reduced the risk of pressure injury development stage ≥ 2 whereas the turning interval of 2 h emerged as non-significant.

#### 3.4.2. Domain 2: Susceptibility and Tolerance of the Individual

Direct causal factors on pressure injury development are both skin/pressure injury status and poor perfusion. Contributing key indirect factors that can influence the aforementioned causal factors are included here, e.g., diabetes, poor nutrition, and health condition or illnesses (Appendix A).

Age

Researchers of 41 studies analysed age as an independent predictor in their multivariate analysis [19,21,22,23,24,27,28,31,32,33,34,35,36,37,39,40,41,43,44,46,48,49,50,51,52,53,54,55,56,58,59,61,62,63,67,72,75,77,82,83,84]. In over half of them (60%) increasing age resulted in a higher risk of pressure injury development. The range of low to high risk of bias studies was bigger for stage ≥ 1 pressure injuries than for stage ≥ 2 pressure injuries, where only moderate and low risk of bias studies included age related variables.

2.Sex

Sex was analysed in 25 studies, where it emerged as a significant predictor in 11 studies (44%) [26,38,49,50,51,53,55,58,59,67,80]. There were more significant results for male sex as a risk factor than female sex.

3.Ethnic group

There is no clear evidence for a specific ethnic group as a risk factor for pressure injury development as only two out of six studies could achieve statistical significance [23,26].

4.Laboratory values

Twenty-one studies evaluated laboratory values in their statistical model [19,24,30,36,38,41,48,50,51,52,53,57,61,66,67,69,71,76,79,81,82]. Lower albumin level was predictive of pressure injury development for stage ≥ 2 pressure injuries in one low [66] and for stage ≥ 1 in three high risk of bias studies [36,76,79], while three moderate [24,61,81] and one high risk of bias [69] studies reported non-significance. Hemoglobin and anemia related variables emerged as significant predictors in five studies (four moderate [48,50,51,61] and one high risk of bias study [71]), with lower levels of hemoglobin increasing the risk of pressure injury development. Six other studies, among them one low [82] and three moderate risk of bias studies [30,38,81], showed no significant results.

5.Nutrition

Fifteen out of 37 nutrition related variables that were studied in 24 studies [20,21,24,26,28,34,40,42,43,47,49,51,53,54,55,61,64,66,75,76,77,79,82,84] emerged as significant predictors of pressure injury development. Significant predictors were related to malnutrition.

6.Body temperature

High body temperature independently predicted pressure injury development in three studies [24,45,71], whereas four studies showed no significant result in their statistical analysis [29,41,53,84].

7.Health status

The American Society of Anesthesiologists’ (ASA) Classification of Physical Health was analysed in three studies [50,62,65] and higher classification emerged as an independent risk factor in multivariate analysis whereas the Acute Physiology and Chronic Health Evaluation (APACHE) did not reach significance in three other studies, among them two low [37,54] and one moderate risk of bias study [81].

8.Diagnosis

Various diagnoses were entered into multivariate analysis in 13 studies [21,36,39,41,50,52,54,55,57,62,64,67,75]. Out of the 27 analysed variables, 11 emerged as independent predictors. Cancer was predictive of pressure injury development stage ≥ 1 in one low risk of bias study [21] but emerged as non-significant in two high risk of bias studies [36,64]. Other significant predictors for pressure injury development stage ≥ 1 include urinary tract infection from one low risk of bias study [57], moderate/severe traumatic brain injury as assessed by the Glasgow Coma Scale in one moderate risk of bias study [39] and spinal cord injury as well as amyotrophic lateral sclerosis in one high risk of bias study [52]. Mixed results were achieved with renal failure and stroke.

9.Subdomain Skin/pressure injury status

Pressure injury stage 1

There is evidence that pressure injury stage 1 is predictive of pressure injury development stage ≥ 2 as there are two low [66,82] and two moderate risk of bias studies [19,20] where it independently predicted pressure injury stage ≥ 2.

Existing/previous pressure injury

Four studies [22,57,60,77] revealed a significant influence of existing pressure injury as a predictor for pressure injury development while previous pressure injuries could not reach significance in any study.

General skin status

Seven studies analysed 17 variables related to general skin status. Dry skin yielded a significant result in one low risk of bias [77] and one moderate risk of bias study [19] while it reached non-significance in one moderate risk of bias study [20]. Conflicting results were also present with edema. Baseline skin trauma in one low risk of bias [82] and mottled skin, reddened skin, and centralized circulation in one moderate risk of bias study [53] were predictive of stage ≥ 2 pressure injuries.

Moisture

Twenty moisture-related variables were included in multivariate analysis in 14 studies [20,28,33,40,46,51,53,61,69,70,73,74,77,84]. Most of these variables were the moisture subscale of the Braden scale. Only eight out of the included 20 variables (40%) emerged as independent risk factors of pressure injury development.

Incontinence

This subdomain includes urinary incontinence, fecal incontinence, and dual incontinence. Only four out of 12 variables were significant predictors of stage ≥ 2 pressure injuries.

10.Subdomain poor perfusion

Diagnosis related to oxygenation and/or perfusion

Diabetes achieved more consistent significant results for pressure injury stage ≥ 2 development. Cardiovascular disease or instability, pneumonia and respiratory disease were reliably predicting pressure injury development, too.

Oxygenation/ventilation

Seven out of 17 variables related to oxygenation and ventilation emerged as significant predictors of pressure injury development.

Vasopressor

Vasopressors were analysed in 10 studies [39,42,53,54,55,56,62,64,68,72]. One low risk of bias study showed that norepinephrine was predictive of deeper pressure injuries than on superficial pressure injuries [54]. Noradrenaline use was significant in one moderate risk of bias study [39]. Overall, vasopressors lack the evidence for significant influence on pressure injuries.

Blood pressure

Hypotension was predictive of pressure injury stage ≥ 1 and ≥ 2 in one low [55] and one moderate risk of bias study [84], respectively, whereas hypertension yielded more non-significant results. Diastolic and systolic blood pressure received mixed results as well.

#### 3.4.3. Additional Domain: Diagnosis and Treatment

An additional domain with variables related to diagnosis or treatment during hospitalization is presented here (Appendix A).

Admission Type

Nine studies analysed the type of admission as a potential risk factor for pressure injury development [41,42,61,62,67,71,75,81,82]. Overall, mixed results were reported as 36% of included variables achieved significance.

2.Length of stay

Length of stay in ICU was predictive of pressure injury development in three low [54,55,72] studies and one moderate risk of bias study [48] but reached non-significance in one low [55] and four moderate risk of bias studies [28,42,50,53]. Duration in other hospital wards showed mixed results as well.

3.Medication

Eight studies included medication in their statistical analysis [26,34,42,51,53,59,64,84]. The overall evidence does not imply a significant connection since only two high risk of bias studies predicted pressure injury development [34,64] whereas several low and moderate risk of bias studies did not.

4.Risk assessment

Braden, Norton, and Waterlow scales were included in multivariate analysis. The total score of the Braden scale achieved more consistent significant results for pressure injuries stage ≥ 1 than stage ≥ 2 while the Norton and Waterlow scale showed no such connection to pressure injury development.

5.Nursing/treatment

Variables revolving around nursing or treatment of patients were analysed in 14 studies [21,32,35,43,44,48,51,53,55,57,59,64,66,72]. Mixed results were achieved with urinary catheters while hemodialysis emerged significant in two studies for pressure injury stage ≥ 1.

## 4. Discussion

After screening 11,565 studies and selecting studies based on sound and rigorous methodology, an updated narrative synthesis of risk factors with 67 included studies was conducted and domains and subdomains oriented according to the conceptual framework of Coleman et al. [6] were created. Overall, the narrative synthesis yielded one subdomain that reliably predicted pressure injury development for stage ≥ 2 which is pressure injury stage 1/non-blanchable erythema. This finding fits with the potential progression and worsening of pressure injuries [87] though the emphasis is on a careful and thorough assessment as there has been concerns regarding its identification. Non-blanchable erythema usually relies on visual assessment through, for example, the use of light finger pressure or transparent plastic films [88]. These assessments are prone to errors due to their subjective nature but there is recent meta-analytic evidence for the effectiveness of specific training programs where overall knowledge about pressure injuries and the visual discrimination ability of nurses increased [89]. Indeed, adequate assessments are key for timely intervention measures, which further highlights the importance of highly educated and qualified nurses or health workers to identify pressure injuries and to distinguish between the stages.

Mechanical boundary conditions

For body mass index, individuals with a BMI < 18.5, classified as underweight according to the World Health Organization, are at greater odds developing pressure injuries than those with higher BMI. These results are in line with common etiological models where pressure injuries are most likely developed on skin areas with a bony prominence as the tissues around them are easily strained and stretched by external forces and are, in turn, susceptible to ischemia and necrosis [90]. Additionally, another study analysed smaller body mass which increased the risk of pressure injury stage ≥ 1 development [83]. Decreased body weight in a fourth study, in this case <58 kg, which represents the lower quartile of baseline weight in this specific study sample, was also predictive of stage ≥ 2 pressure injury [19]. There were, however, other studies where BMI classifications or weight did not reach significance [26,34,49,53].

There is some evidence that overlay mattresses or protective support surfaces may prevent further development of pressure injuries. Preventive transfers on special mattresses, heel elevation, a multi-layer foam applied to the sacrum area, and alternating pressure air overlays successfully decreased the risk of pressure injury development in the respective area [28,37,78,80], displaying the potential of specific overlays for pressure injury prevention. Further research is still needed, though, as there are studies which could not prove the usefulness of protective overlays [20,42,82].

2.Susceptibility and tolerance of the individual

Demographic characteristics, e.g., age, showed more significant results as an independent risk factor for pressure injuries than non-significant, whereas for sex and ethnic groups, more mixed results have been achieved, with ethnic groups suffering from a low amount of studies to base these results on. A more thorough view on age and sex is detailed in a separate meta-analysis.

Three moderate risk of bias studies analysed anemia as a risk factor. Two studies imply that patients with anemia have increased risk of developing pressure injuries whereas the third study showed no significant connection. Hemoglobin is considered to be the most reliable method to assess anemia [91]. With low hemoglobin concentration, oxygen cannot be carried to organs and tissues effectively, which can negatively affect wound healing [92]. The majority of studies which analysed hemoglobin itself, however, reached non-significance in their statistical analysis. More data on anemia and/or hemoglobin are needed to make a conclusive and cohesive statement regarding their connection to pressure injury development.

For surgery patients, there is evidence that the ASA classification (American Society of Anesthesiologists) might be a good predictor for pressure injury development. Higher scores indicate a worse physical condition of a surgical patients, ranging from 1 (healthy) to 6 (brain-dead) [93]. Three studies analysed the ASA classification and found significant connections between higher ASA scores and pressure injury development (score ≥ 3 [50], scores 3, 4 vs. 1, 2 [65] and scores 4, 5 vs. 1, 2, 3 [62]). These results suggest that patients with severe systemic diseases are at greater risk of developing pressure injuries. Since the score alone does not represent a single illness or condition but rather a conglomerate of potential several diseases, surgery patients prone to severe diseases should be treated very thoroughly.

Concerning past pressure injuries, more evidence can be found for present or existing pressure injuries as a risk factor for the development of new pressure injuries compared to previous pressure injuries [22,57,60,77] which fits with the evidence of non-blanchable erythema as a risk factor. Patients who are admitted with pressure injury are therefore particularly at risk at developing new pressure injuries or worsening their already existing ones if left untreated.

Diabetes achieved more consistent results with stage ≥ 2 than stage ≥ 1 pressure injuries which suggests a bigger influence of diabetes on deeper pressure injuries than on superficial ones. Mainly low risk of bias studies achieved overall significant results for diabetes as a risk factor for pressure injuries stage ≥ 2 whereas a mixture of low to high risk of bias studies showed conflicting results for stage ≥ 1. Two studies analysed hip-fractured patients within their sample and those with diabetes were at higher risk developing pressure injuries [36,65]. In general, diabetes is linked to a variety of health deficiencies, including increased risk for vascular diseases, anxiety disorders, and cancer [94,95,96,97]. It is therefore of the utmost importance to consider diabetes as a risk factor for pressure injury development and to initiate proper interventions if necessary. In that way, further health complications can be avoided.

Regarding blood pressure, there is more evidence for the connection between hypotension and pressure injury development than with hypertension [55,68,84]. Hypotension affects the oxygenation of tissues which can result in hampering wound healing. Although hypotension is generally viewed as not threatening as hypertension, concerning pressure injury development, it should be considered thoroughly and not be neglected or viewed as unimportant.

3.Diagnosis and Treatment

The data revolving around length of stay in ICU are mixed, though more low risk of bias reported significant connections between longer length of stay in ICU and pressure injury development [54,55,72]. A longer stay in ICU relates to limited movement and longer duration in restricted positions which might benefit the development of pressure injuries. As with other risk factors, more extensive research is needed to flesh out the data.

Among the risk assessment scales used to predict the risk of developing pressure injuries, the Braden scale was analysed far more than the Waterlow or Norton scale and provided more significant results than the others. Further details for the Braden scale are outlined in a separate meta-analysis. Only two studies included the Waterlow scale in their multivariate analysis and five studies analysed the Norton scale, the former achieving significant results in one high risk of bias study [76] and non-significant results in one moderate risk of bias study [53] and the latter showcasing non-significant results in the majority of studies [29,40,41,77], which refer mostly to the subscale “physical condition” of the Norton scale. As both scales are also widely used, more data regarding their total score and their respective subscales are needed.

### Limitations

Despite methodological strengths and careful evaluation, overall limitations stem mostly from the quality of the included studies. Thus, 45% were assessed as moderate and 23% were assessed as high risk of bias with the QUIPS tool [10]. The domains “outcome measurement” and “statistical analysis and reporting” had the highest share of high risk of bias assessments.

“Outcome measurement” was appraised based on two factors: how were pressure injuries defined and staged as the main outcome and who was responsible for identifying and staging pressure injuries. Most studies referred to the respective classification and staging system of the EPUAP, NPIAP and PPPIA while others used the definition of ICD 9 or ICD 10 codes. A few did not state any source of pressure injury definition which makes it difficult to evaluate if pressure injuries are correctly defined in the given study. Another aspect that was not covered in most studies refers to the assessment of pressure injuries since it was unclear if the responsible person is trained and skilled enough to determine pressure injuries correctly. Ideally, they are explicitly trained for the purpose of identifying and staging pressure injuries through specific training. Nurses or health workers are suited for this process. However, some studies failed to deliver sufficient information about it which limits comparability between studies.

The subdomain “statistical analysis and reporting” was rated based on the amount of given information regarding analytic approach, statistical model building and reporting of results. Another important point is the “ten in one” rule of thumb, which suggests enough events per predictor included in multivariate analysis, in this case, at least 10 newly developed pressure injuries for one included predictor. Many studies failed to meet this condition which led to poor accuracy and precision of regression models, as shown by the simulation study by Peduzzi et al. [17]. Future research should consider both the accurate and professional assessment of pressure injuries by trained and skilled personnel and the statistical and methodological inclusion of an adequate number of newly developed pressure injuries in multivariate models as both are very important and key to correctly determine potential risk factors.

Further limitations can be derived from the selection of inclusion and exclusion criteria which are based on the widely used PICOS framework [8]. As previously mentioned, the “comparator/control” domain was not included due to the lack of control groups. The Joanna Briggs Institute (JBI) proposes similar selection criteria for systematic reviews of etiology and risk, specifically [98], which could have increased the accuracy of included studies in this review.

Additionally, the inclusion of different study designs in narrative synthesis, while comprehensive, does pose limitations regarding the comparability among studies.

## 5. Conclusions

Overall, the evidence brought by narrative synthesis is characterized by mixed results which was true with previous reviews regarding risk factors of pressure injury development and underlines the interplay of various risk factors. Even with the inclusion of newer studies in the past years, it is evident that heterogeneity revolving around the assessment and operationalization of risk factors, pressure injury assessment and grading, study population, and statistical analysis impede the successful synthesis of data. The need for a more unified and condensed approach to risk factor research regarding pressure injury development is necessary to reach a consensus on different predictors.

In the sense of primary prevention, susceptible individuals, e.g., older patients admitted to hospitals, benefit from healthy nutrition to prevent a worsening of their health condition. The optimization of mechanical boundary conditions by special mattresses, heel elevation, or multi-layer foam can support prevention in nursing practice. The close mutual communication between physicians and nursing staff is vital to ensure the ongoing awareness of decubitus-relevant illnesses or symptoms, such as diabetes, hypoalbuminemia, anemia, or hypotension. Physiotherapists may counteract loss of activity, which leads to immobilization.

Patients who are admitted with already existing pressure injury are at risk at developing new pressure injuries or worsening the already existing ones. Therefore, nurses and health personnel should be vigilant about non-blanchable erythema and the correct assignment of pressure injury stages when assessing the skin status of new patients. As non-blanchable erythema is prone to subjective perception, specific training in identifying those should be considered if necessary to increase the accuracy and validity of skin assessments.

In summary, the evidence-based knowledge of risk factors for pressure injury is vital not only for nursing practice, but for all health care professionals, because optimal prevention or treatment can only be ensured by a close and trustworthy collaboration.

## Figures and Tables

**Figure 1 ijerph-19-00761-f001:**
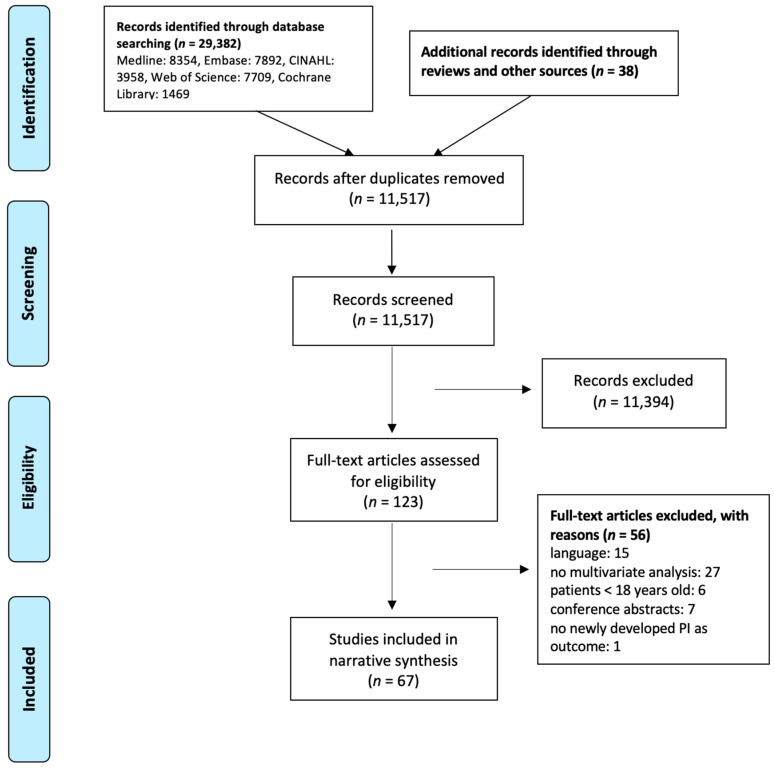
Flow chart decision process.

**Figure 2 ijerph-19-00761-f002:**
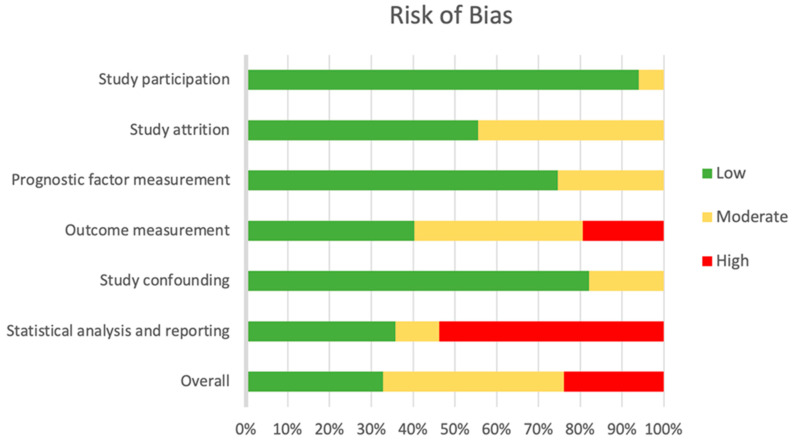
Risk of bias graph.

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
