# Peer review of "Risk Factors for Pressure Injuries in Adult Patients: A Narrative Synthesis"

_ijerph, 2022, doi:10.3390/ijerph19020761_

Round 1

Reviewer 1 Report

This is a very interesting and long-awaited review of pressure injury factors, I am very happy to be able to review it. Please pay attention to the nomenclature related to the nomenclature from 2016, the term pressure injury should be used, it is an EPUAP / NPIAP recommendation.
  1. Edsberg L.E., Black J.M., Goldberg M., McNichol L., Moore L., Sieggreen M.: Revised National Pressure Ulcer Advisory Panel Pressure Injury Staging System Revised Pressure Injury Staging System. J Wound Ostomy Continence Nurs. 2016;43(6):585-597
  2. Kottner J., Cuddigan J., Carville K.,( ), et al.: Prevention and Treatment of Pressure Ulcers/ Injuries Clinical practice Guidelinw; The International Guideline 2019; European Pressure Advidory Panel, National Pressure Injury Advisory Panel and Pan Pacific Pressure Injury Alliance, 2019
Did you use the phrase "pressure injury" in the search results ??   The whole analysis is legible, understandable and presented in an appetizing way, it is perfectly readable. Please note that the conclusions are too extensive and the citation of the authors in this case is not desirable, I would suggest to unify it and clarify it  

Reviewer 2 Report

Thank you for being able to review this manuscript. Here are some areas for improvement of the manuscript
The objective of the study does not correspond to a research objective, both in the abstract and in the introduction ("Evidence from the past decade has not been systematically analysed yet" ).
The introduction is confusing and disorganized. It begins by justifying the objective of the study and ends with a description of the risk factors for pressure ulcers. A good justification of the study is needed.
Methodology: 
Line 65-67. If the studies do not have control groups, why are RCTs included in the inclusion criteria? It does not make sense.
Inclusion criteria: Why studies of more than 200 subjects? Including studies of different designs is a major bias for the study, which does not appear in the study limitations. 
"Any kind of treatment or intervention regarding pressure ulcer prevention". Should be more specific.
Regarding the exclusion criteria, it is repeating the opposite of the inclusion criteria. The search strategy should be considered.
The search strategy, what about MeSH terms, were they used?
Risk of Bias. I advise the author to simplify this section because it leads to confusion for the readers.
Statistical analysis and reporting. Has statistical analysis really been performed?
Adequate discussion and conclussion 

Round 2

Reviewer 2 Report

The authors have responded correctly to the suggestions made. Therefore, I consider that the manuscript can be accepted.